# Oncology-Led Early Identification of Nutritional Risk: A Pragmatic, Evidence-Based Protocol (PRONTO)

**DOI:** 10.3390/cancers15020380

**Published:** 2023-01-06

**Authors:** Maurizio Muscaritoli, Gil Bar-Sela, Nicolo Matteo Luca Battisti, Borislav Belev, Jorge Contreras-Martínez, Enrico Cortesi, Ione de Brito-Ashurst, Carla M. Prado, Paula Ravasco, Suayib Yalcin

**Affiliations:** 1Department of Translational and Precision Medicine, Sapienza University of Rome, 00185 Rome, Italy; 2Oncology Department, Emek Medical Center, Afula 1834111, Israel; gil_ba@clalit.org.il; 3The Royal Marsden NHS Foundation Trust, London SW3 6JJ, UK; nicolo.battisti@rmh.nhs.uk (N.M.L.B.); ione.ashurst@rmh.nhs.uk (I.d.B.-A.); 4Breast Cancer Research Division, The Institute of Cancer Research, London SW3 6JJ, UK; 5Clinical Hospital Center Zagreb, School of Medicine, University of Zagreb, 10000 Zagreb, Croatia; borislavbelev@gmail.com; 6Hospital Regional de Malaga, 29010 Malaga, Spain; jorgecontrerasmartinez@gmail.com; 7Department of Radiological, Oncological and Pathological Sciences, Sapienza University of Rome, 00185 Rome, Italy; enrico.cortesi@uniroma1.it; 8Department of Agricultural, Food and Nutritional Science, University of Alberta, Edmonton, AB T6G 2R3, Canada; carla.prado@ualberta.ca; 9Faculty of Medicine and Centre for Interdisciplinary Research in Health (CIIS-UCP), Universidade Católica Portuguesa, 1649-023 Lisbon, Portugal; pravasco@ucp.pt; 10Centre for Interdisciplinary Research Egas Moniz (CiiEM), Instituto Universitário Egas Moniz, 2829-511 Almada, Portugal; 11Department of Medical Oncology, Institute of Cancer, Hacettepe University, Ankara 06800, Turkey; suayibyalcin@gmail.com

**Keywords:** nutrition, malnutrition, strength, mobility, cachexia, sarcopenia, cancer, antineoplastic therapy, risk identification, protocol, PRONTO

## Abstract

**Simple Summary:**

Early identification of patients on antineoplastic therapy who are at risk for or already malnourished is critical for optimizing treatment success. Malnourished patients are at increased risk for being unable to tolerate the most effective ‘level’ and ‘duration’ of treatment, with grave implications for both the short- (during treatment) and long-term outcomes. Herein, we provide a practical PROtocol for NuTritional risk in Oncology (PRONTO) to enable oncologists to identify patients with or at risk of malnutrition for further evaluation and follow-up with members of the multidisciplinary care team (MDT). Additional guidance is included on the oncologist-led provision of nutritional support if referral to a dietary service is not available.

**Abstract:**

Nutritional issues, including malnutrition, low muscle mass, sarcopenia (i.e., low muscle mass and strength), and cachexia (i.e., weight loss characterized by a continuous decline in skeletal muscle mass, with or without fat loss), are commonly experienced by patients with cancer at all stages of disease. Cancer cachexia may be associated with poor nutritional status and can compromise a patient’s ability to tolerate antineoplastic therapy, increase the likelihood of post-surgical complications, and impact long-term outcomes including survival, quality of life, and function. One of the primary nutritional problems these patients experience is malnutrition, of which muscle depletion represents a clinically relevant feature. There have been recent calls for nutritional screening, assessment, treatment, and monitoring as a consistent component of care for all patients diagnosed with cancer. To achieve this, there is a need for a standardized approach to enable oncologists to identify patients commencing and undergoing antineoplastic therapy who are or who may be at risk of malnutrition and/or muscle depletion. This approach should not replace existing tools used in the dietitian’s role, but rather give the oncologist a simple nutritional protocol for optimization of the patient care pathway where this is needed. Given the considerable time constraints in day-to-day oncology practice, any such approach must be simple and quick to implement so that oncologists can flag individual patients for further evaluation and follow-up with appropriate members of the multidisciplinary care team. To enable the rapid and routine identification of patients with or at risk of malnutrition and/or muscle depletion, an expert panel of nutrition specialists and practicing oncologists developed the PROtocol for NuTritional risk in Oncology (PRONTO). The protocol enables the rapid identification of patients with or at risk of malnutrition and/or muscle depletion and provides guidance on next steps. The protocol is adaptable to multiple settings and countries, which makes implementation feasible by oncologists and may optimize patient outcomes. We advise the use of this protocol in countries/clinical scenarios where a specialized approach to nutrition assessment and care is not available.

## 1. Introduction

Patients with advanced (metastatic) cancer often experience nutritional problems, including malnutrition, low muscle mass, sarcopenia (i.e., loss of muscle mass and function), and cachexia (i.e., a form of disease-related malnutrition characterized by weight loss and a continuous decline in skeletal muscle mass, with or without fat loss) [1,2,3]. However, there is accumulating evidence that patients with early-stage (non-metastatic) disease also experience nutritional issues [4], including weight loss and muscle loss, which may impact long-term outcomes. Studies have highlighted the prevalence of malnutrition and/or muscle depletion at the time of cancer diagnosis [4,5,6,7].

In the PreMiO study among treatment-naïve patients, 2.7% of those with Stage 1 disease were malnourished at the time of diagnosis, increasing to 15.2% of those with Stage 4 disease at the time of diagnosis [4]. The prevalence of an increased risk of malnutrition at diagnosis was 20.1% of those with Stage 1 disease, increasing to 42.7% of those with Stage 4 disease [4]. Weight loss is a defining criterion of malnutrition. It occurs regardless of cancer type or stage and is associated with shorter survival. Recently, a large study including 12,253 patients at risk for cancer-associated weight loss clearly established a correlation between reduced food intake and inflammation and survival [8].

The progressive loss of skeletal muscle mass is probably the most clinically relevant feature of disease-related (and cancer-related) malnutrition [9]. Low muscle mass has been shown to increase the risk of dose-limiting toxicity during systemic antineoplastic therapy [10,11,12,13,14,15,16,17,18,19]. Furthermore, by preventing the delivery of an optimal regimen, muscle depletion can directly compromise the efficacy and outcomes of anticancer treatments [15,20,21].

For these reasons, nutritional evaluation should be undertaken for all patients diagnosed with cancer regardless of their disease stage and should be monitored throughout their treatment journey and beyond, with the aim of optimizing outcomes while reducing treatment-related side effects. Indeed, early recognition of nutritional issues is key for improving quality of life and avoiding patients becoming unfit for therapy as their treatment progresses [22,23]. Optimal nutritional status improves a patient’s ability to tolerate antineoplastic therapy, is associated with a reduction in post-surgical complications following surgery, and has a positive impact on long-term outcomes, including survival [24,25,26,27,28].

In many countries, dietitians form part of the MDT for cancer care pathways and utilize available tools for assessment of the presence of malnutrition and sarcopenia (see Appendix A). However, healthcare services are frequently underfunded and understaffed, with long waiting times for referrals, and in some cases, there is no access to dietitian services at all [29]. As such, there is a clear need for a standardized approach to evaluate and monitor nutritional status by the oncologist or physician leading patient care, in order to maximize treatment efficacy and minimize the risk of toxicity. There have been recent calls to establish nutritional screening, assessment, treatment, and monitoring as a core and standard component of care for all patients diagnosed with cancer [30,31,32,33]. The challenge then is to establish when and by whom this critical task should be undertaken and facilitate the process in a way that is feasible and effective within a busy oncology practice [33].

Experts agreed that it is essential to identify malnourished patients and those who are at risk of malnutrition, including those experiencing low muscle mass. Moreover, identification helps patients better understand the importance of nutrition and nutritional status during antineoplastic therapy and enables support. Malnourished patients have a reduced chance of benefiting from their treatment [26,34]. These are critical considerations in all age groups, when malnutrition and/or muscle depletion, which may lead to sarcopenia and cancer cachexia, correlate not only with reduced efficacy and higher toxicity of anticancer treatments, but also with a worse quality of life, functional impairment, and increased mortality [35,36].

Herein, we provide oncologists with a practical protocol—PROtocol for NuTritional risk in Oncology (PRONTO)—that can be used as a tool for the early identification of patients at risk of malnutrition and/or muscle depletion for further evaluation and follow-up with members of the multidisciplinary care team (MDT). The PRONTO protocol was designed with the intent to be suitable for most oncology settings, where nutritional care can be optimized for patients with cancer. The simple protocol can be followed when patients are about to start antineoplastic therapy, regardless of their disease type or stage, and is simple and rapid enough to be used for regular monitoring to identify emergent nutritional issues at any time during their treatment course. The protocol is not intended to replace existing guidance for the comprehensive evaluation of nutritional status or clinical tools intended to diagnose malnutrition, such as the recently proposed GLIM consensus criteria [37], but rather to ensure that patients requiring such evaluation and support are identified promptly. Additional guidance is provided for the management of at-risk patients where referral to dedicated nutritional services is not available and the nutritional status of the patient must be managed within the oncology department.

## 2. Methods

A cohort of international experts in cancer-related nutrition and practicing oncologists was convened in October 2021. The expert team met twice in October 2021, with additional discussions occurring via email. The first meeting identified the challenges and potential barriers perceived by practicing oncologists to the implementation of nutritional evaluation in the context of an oncology consultation. The second meeting identified the key concepts and framework of a protocol that would be suitable for use by medical or radiation oncologists in routine clinical practice as well as by appropriate members of the oncology service, such as nurses or, where available, dietitians and nutritionists.

Validated tools to determine the nutritional risk of patients were identified and considered for their suitability as part of a rapid checkpoint protocol in routine oncology practice (please see Appendix A) [26,37,38,39,40,41,42,43,44]. In the absence of widely available tools to identify low muscularity, alternative approaches were discussed, such as tools to assess strength/mobility, such as handgrip dynamometry and the ‘chair stand’ test. As low muscle mass is a component of the sarcopenia definition, and the majority of patients with cancer are older adults, the group considered sarcopenia screening tools may be used as surrogates.

The expert panel recognized the availability of a number of validated screening tools; however, oncologists in the group raised the global inconsistencies with access to dietitians and nutritional services for referral as well as time limitations for in-depth screening during patient consultations, should this fall to the oncologist. Ideally, patients with or at risk of malnutrition or low muscle strength/mobility will be referred directly and immediately to a dietitian/clinical nutrition specialist who is an active part of the multidisciplinary oncology team and well-versed in nutritional assessment using existing validated tools. Indeed, in some countries, nutritional intervention can only be undertaken by a dietitian. Where this is not possible, the group concluded that having a simple protocol for oncologists to rapidly check risk of malnutrition and muscle depletion—and quickly take action—would be of most value to ensure the best patient outcomes. Hence, core components of tools were identified to form the basis of the PROtocol for NuTritional risk in Oncology (PRONTO).

## 3. PRONTO: An Evidence-Based Protocol for Early Identification of Nutritional Risk for Patients with Cancer

### 3.1. Identification of Patients with or at Risk of Malnutrition and/or Muscle Depletion by Oncologists

Following review of available tools, a minimum set of informative questions to determine the risk of malnutrition and muscle depletion in patients scheduled to begin antineoplastic therapy and for monitoring any change in patients’ nutritional status prior to and throughout the treatment course was identified.

The expert panel identified three factors as essential components to give a rapid notion of patients’ nutritional and physical status: (1) body weight, (2) appetite and food intake, and (3) strength and mobility (Figure 1). Strength and mobility are used as substitutes (but not as proxies for muscle mass) for the evaluation of muscle mass, for reasons mentioned previously.

Unintentional weight loss can be an early sign of a malignancy and is an established marker of disease-related malnutrition [37]. Indeed, since unintentional weight loss in patients with cancer has been shown to independently predict reduced survival [49], evaluation of the change in body weight was recommended for all patients (regardless of body weight or body mass index) at the time of diagnosis and throughout treatment. ESMO guidelines define loss of or low body mass as non-volitional weight loss >5% in 6 months [30]; therefore, where possible, the expert panel felt it would be appropriate to quantify this measurement in patient discussions and as part of the protocol.

An important contributor to weight loss is reduced appetite and food intake [8,30,45]. A patient’s appetite and food intake should be evaluated in addition to monitoring body weight, as these are potential early indicators of risk for weight loss and malnutrition. Of particular concern is cancer-related anorexia (a reduced desire to eat). Anorexia is common among individuals with cancer and is an established side effect of numerous antineoplastic therapies [4,46].

Notably, cancer-related anorexia may not be the only reason for reduced food intake in newly diagnosed patients or for those undergoing antineoplastic therapy. There may be physical limitations preventing consumption of a normal diet. There may also be factors related to the tumor type affecting their ability to eat or to efficiently utilize nutrients from foods they consume or factors related to their treatment (e.g., surgery for head and neck cancer). Additional barriers may involve oral mucositis and dysgeusia associated with systemic anticancer treatments [50,51]. Early identification of patients experiencing these issues is essential to ensure timely prescription of and optimal adherence to nutritional supplementation and to achieve their daily nutritional requirements.

Reductions in patient-perceived strength and in their general mobility may be related to muscle depletion, such as their ability to carry out ordinary daily activities (i.e., open bottles, stand up from a chair, climb up the stairs, or lift heavy objects) [30,45]. Low muscle mass is common in patients with cancer, and >50% of newly diagnosed patients with cancer exhibit some degree of muscle depletion [52,53]. Importantly, muscle depletion may occur independently of body weight loss or a drop in body mass index. For this reason, changes in muscle mass should be identified in addition to changes in body weight [54], especially as muscle depletion can be hidden in patients who have excess body weight (i.e., overweight or obese) at the time of diagnosis [47,55].

Figure 1 outlines the evidence-based protocol for rapid identification of patients with or at risk of malnutrition and/or muscle depletion by their oncologist (PRONTO: PROtocol for NuTritional risk in Oncology). The rapid identification of such patients enables their prompt referral for additional assessment. It is not intended to replace validated screening tools but to facilitate early recognition of nutritional risk by the oncologist within the time constraints of routine clinical practice.

### 3.2. When Should Nutritional Checks Be Undertaken in Patient Consultations and by Whom?

Certain tumor types, such as pancreatic, upper gastrointestinal, head and neck, or of the respiratory tract, are associated with a higher risk of malnutrition and low muscle mass compared with other tumor types. In a French study of adults with cancer, the prevalence of malnutrition ranged from 18.3% in breast cancer to 49.5% in upper digestive cancer [56]. The expert panel agreed that there was clear evidence for the need for identification of low muscle mass, as patients with certain tumor types and those receiving certain systemic therapies are at risk for muscle depletion (Table 1).

An international consensus has recommended nutritional screening for older patients with cancer being considered for chemotherapy [32]. However, most patients with cancer are at increased risk of malnutrition and low muscle mass regardless of age and tumor type [4]. For this reason, the expert panel agreed that early identification of nutritional risk should be undertaken at the time of initial diagnosis [22,30,45], regardless of age, disease stage, or cancer type (Figure 1). As such, checkpoints for identification of nutritional risk should be undertaken at any scheduled or unscheduled medical oncology/radiation oncology visit. In addition, the expert panel recommended that the checkpoints for identification of nutritional risk should be performed at the time of disease recurrence, and whenever a change of treatment is being considered, prior to and following surgery (particularly head–neck, esophagectomy, gastrectomy, small bowel resection, or pancreatectomy).

The expert panel advocates that identification of the initial nutritional risk of a patient should be undertaken by the medical or radiation oncologist and/or the oncology nurse, as well as the dietitian (when available) or other attending healthcare professional leading patient care. Some specialist centers may have a nutritional support service to which patients are automatically referred for nutritional evaluation and support at the time of diagnosis. However, for most patients, their primary interaction will be with their treating physician, and this is the reason for the recommendation. Ideally, patients identified at regular checkpoints and via monitoring of nutritional status during antineoplastic treatment can then be referred for detailed nutritional assessment by nutrition experts. There was a full consensus that it is not the role of medical oncology specialists to undertake full nutritional assessments, but if a referral is not possible, early identification of malnutrition and muscle depletion should be a key point for management by members of the multidisciplinary team.

### 3.3. How Should Malnutrition and Muscle Depletion Be Identified in Patients with Cancer?

Having established the three key factors to identify patients with or at risk of malnutrition and/or low strength/mobility (in lieu of muscle mass), the expert panel identified a core set of questions used in established, validated screening tools (Figure 1). Notably, we will hereby refer to strength/mobility when discussing the specific protocol criteria. The expert panel agreed that a minimum number of essential questions should be included, as the intention would be to identify patients requiring referral for specialist nutritional assessment and support.

To assess changes in body weight, it was agreed to ask patients about usual body weight as a baseline, in addition to measuring their current weight, and to ask specifically about unintentional weight loss in the previous 3–6 months. It is especially important to recognize the pace of weight loss early in the course of disease and to highlight that many patients will have lost appreciable weight prior to presenting to healthcare [37]. For patients who may be less aware of their weight or any changes, the experts proposed asking about any recent changes in how well clothes, jewelry, dentures, or belts fit.

One single question focused on appetite and food was agreed as sufficient, with patients being asked whether they had been eating less than usual in the last week or since the last consultation. This simple question could be used to open a discussion about any symptoms of the disease or side effects of treatments that might be impacting a patient’s appetite or food intake, for which intervention may be required.

Finally, one single question was considered sufficient for the purposes of strength/mobility, which may be associated with decreased muscle mass, by asking patients if they perceived any loss of strength or weakness in their daily lives. Asking patients about their ability to perform simple tasks would be useful, as would objective tests such as handgrip strength, the ‘chair stand’ test, or the ‘timed up and go’ test, should consultation time allow. Notably, these latter evaluations were not considered mandatory by the expert panel to include in a checkpoint for identification of risk.

Based on the above, the expert panel agreed that three questions should be asked by the oncologist or physician leading patient care for early identification of nutritional risk:For the evaluation of weight loss: “Have you unintentionally lost weight (5% to 10% or more) in the last 3–6 months/since our last consultation?”For the evaluation of appetite and food intake: “Have you been eating less than usual in the last week/since our last consultation?”For the evaluation of strength and mobility: “Have you lost strength or do you feel weaker than usual/since our last consultation?”

### 3.4. Managing the Patient with or at Risk of Malnutrition and/or Muscle Depletion

Identification of patients with or at risk of malnutrition and/or muscle depletion should ideally be followed by referral to appropriate members of the multidisciplinary care team for full nutrition and strength/mobility assessments to guide a tailored therapy intervention plan. However, as the availability of professionals and time limitations might hinder such actions, the protocol also provides guidance on how to manage and monitor patients during treatment (Figure 1). The ESPEN recommendations for actions [45] also link to further evidence-based nutrition resources for both healthcare professionals and patients.

Patients whose weight loss can be documented as <5% of their usual body weight over the last 3–6 months should be recommended to monitor their weight. Additionally, these patients should be provided with basic dietary counselling to maintain or improve nutrient intake during treatment. Physical activity should be encouraged and has a range of benefits, including improved muscle strength, reduction in fatigue and anxiety, and improved quality of life [45]. Patients reporting no change in strength or mobility should be advised to monitor their activity levels and should be encouraged to engage in regular physical activity to reduce the risk of atrophy [45].

Early nutritional support for patients identified with or at risk of malnutrition or muscle depletion has the potential to reduce the possibility of therapy-threatening adverse events and to optimize the likelihood of treatment success and long-term survival [33,52,53]. Early nutritional intervention may be associated with improved outcomes and a better quality of life, including emotional and psychological status [33,52,53]. Additional studies are emerging that highlight the benefits of early and prospective nutritional management during systemic antineoplastic therapy [24,64,65]. Specifically, studies have shown the benefits of oral nutritional supplements and enteral nutrition for patients undergoing antineoplastic therapy [24,66,67,68,69,70,71,72,73].

Where a dietitian/clinical nutrition specialist is an active part of the multidisciplinary oncology team, best practice sees patients with or at risk of malnutrition or low strength/mobility referred directly and immediately for further assessment. However, the expert panel recognized that referral for a dietitian/clinical nutrition specialist is not always available, and the treating oncologist may be required to advise and ensure that patients have access to nutritional supplementation, as required.

Detailed guidance on nutritional support and intervention for patients diagnosed with cancer who are undergoing antineoplastic treatment has been provided by the European Society for Clinical Nutrition and Metabolism (ESPEN) [45] and the European Society for Medical Oncology (ESMO) [30] (Figure 2). These guidelines provide recommendations for total daily energy that should meet the standard daily energy expenditure of healthy adults of between 25 and 30 kcal/kg/day, protein intake of >1 g/kg/day, or, if possible, up to 1.5 g/kg/day, and vitamin and mineral supplementation equal to the recommended daily amounts for healthy individuals. The source of protein should also be considered for optimal muscle anabolism [74]. Reaching caloric requirements might be achieved with the usual recommendations of “little and often” and “fortified foods” or with oral nutritional supplements, which can also improve protein and micronutrient intake. Furthermore, if oral intake remains inadequate to meet requirements despite added oral nutritional supplements, enteral nutrition must be considered. Guidelines also recommend maintenance or an increased level of physical activity in patients with cancer to support muscle mass, strength/mobility, and metabolic pattern (ESPEN) [45]. Physical activity and individualized resistance exercise can support health-related quality of life, self-esteem, as well as a reduction in fatigue and anxiety for patients with cancer and should be encouraged in order to reduce risks of atrophy due to inactivity. (ESPEN) [45].

### 3.5. Implications of Early Identification of Nutritional Status and Patient Risk

For most patients, the focus should be on identifying those at increased risk of malnutrition and/or muscle depletion at the time of diagnosis and monitoring for any change in status throughout treatment. Nutritional support should be valued as an essential component of the holistic management of all patients with cancer undergoing antineoplastic therapy in the same way oncologists routinely evaluate a range of factors, including standard considerations such as blood counts and organ function, to determine whether a patient is fit to commence antineoplastic treatment [75,76]. Where possible, delays in initiating antineoplastic therapy due to poor nutritional status should be avoided via the early identification of nutritional status and patient risk. However, the need to initiate immediate antineoplastic therapy must be considered alongside the risk for treatment-related toxicities and early treatment discontinuation in those whose nutritional status is poor. To prevent this nutritional decline, a standardized approach for identification and monitoring of nutritional risk of patients commencing and undergoing antineoplastic therapy is proposed with a nutrition awareness protocol (PRONTO) that is feasible within the context of a demanding oncology practice.

## 4. Conclusions

Access to nutrition care is regarded as a basic human right [23]. While it is beyond the purview of oncologists to ensure that every patient has access to adequate food, it is incumbent on them to ensure that those patients with or at risk of malnutrition and/or muscle depletion are identified and managed appropriately. Ideally this would be via referral to members of the multidisciplinary team to enable provision of a full nutritional assessment, dietary counselling, and nutritional support, including oral nutritional supplements and enteral and parenteral nutrition where needed, to optimize nutritional status as they undergo antineoplastic therapy. Early identification of patients with cancer at risk for or with malnutrition and/or muscle depletion who would benefit from an optimal nutrient intake is therefore essential. Well-nourished patients are better able to tolerate and complete their antineoplastic therapy and have better outcomes, including improved quality of life and survival. The UN estimates a projected shortfall of 18 million healthcare workers by 2030 [29], which includes dietitians. There is a need for more specialized nutrition professionals, and whilst ideally we would recommend access to a dietitian for all or most patients with cancer, this cannot be fulfilled in many healthcare settings.

The authors intend for the PROtocol for NuTritional risk in Oncology (PRONTO) to be suitable for implementation in most oncology settings to achieve the rapid identification and referral of patients requiring nutritional evaluation and support. Notably, this protocol does not aim to replace nutritional assessment or existing tools used by a dietetic/clinical nutrition service/facility that works closely with the oncology team as part of the multidisciplinary team. However, it intends to be a complementary source of consensus used by oncologists to support the essential early identification of patients at risk of malnutrition and/or muscle depletion and to prevent nutritional decline. Where a dietetic/clinical nutrition service/facility is not available, medical/radiation oncologists and oncology nurses may provide basic nutrition information for patients, including prescribing oral nutritional supplements, as necessary. Identifying patients who are at risk of or with malnutrition and/or muscle depletion should be a core part of the holistic management of patients undergoing antineoplastic therapy from the time of their initial diagnosis and throughout their treatment journey. The PROtocol for NuTritional risk in Oncology (PRONTO) presented here can support this potentially life-saving process within the practical limitations of oncology services. Future studies should validate the feasibility and effectiveness of this tool. Indeed, it is important that the oncology and the nutrition communities act jointly to consider the use of the PRONTO framework both in prospective and retrospective cohort studies as well as clinical trials in order to confirm its relevance for clinical practice, promoting dissemination, validation testing, and feedback.

It is the opinion of the expert panel that the routine use of the criteria proposed in the PRONTO protocol has the potential for significantly improving routine nutritional care in oncology patients with a simple, essential, evidence-based, and inexpensive approach, without negatively impacting the already overburdened medical/radiation oncologist’s daily schedule.

## Figures and Tables

**Figure 1 cancers-15-00380-f001:**
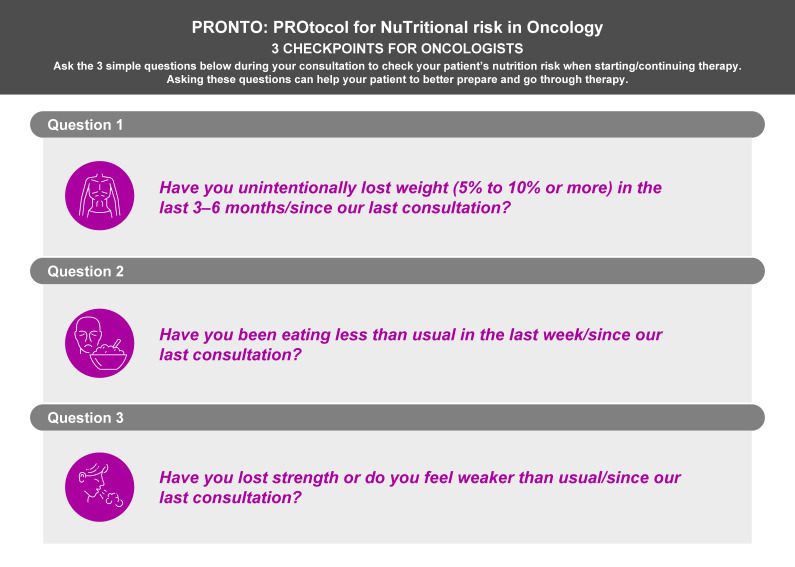
PRONTO: PROtocol for NuTritional risk in Oncology [4,8,26,30,37,45,46,47,48].

**Figure 2 cancers-15-00380-f002:**
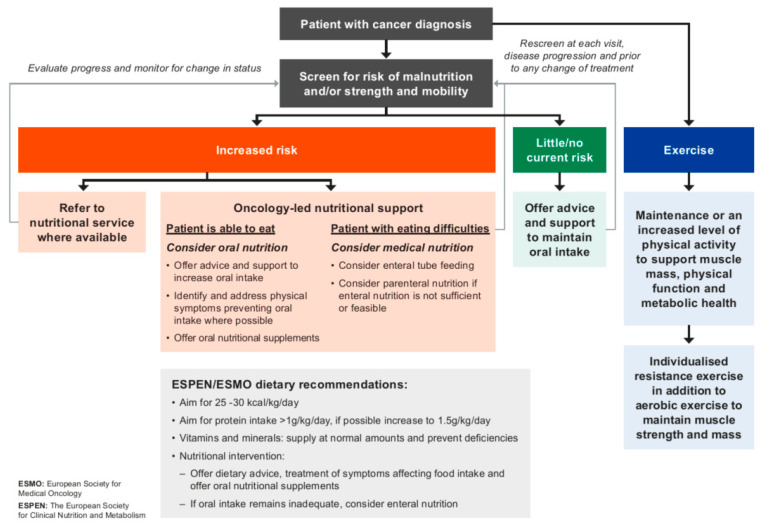
Decision-making algorithm following screening for the risk of malnutrition/loss of muscle mass in patients with a diagnosis of cancer, should no dedicated nutrition service be available. Content adapted and developed from ESMO and ESPEN evidence-based guidelines [30,45].

**Table 1 cancers-15-00380-t001:** Tumor types and/or treatments most closely associated with malnutrition and/or muscle depletion (adapted from Bozzetti 2017 [48]) [48,57,58,59,60,61,62,63].

Tumor types commonly associated with low muscle mass ^a^	Lung Pancreas Colorectal Liver Esophagus Kidney Bladder Breast (metastatic) Non-Hodgkin lymphoma Stomach Genito-urinary
Chemotherapy regimens/drugs associated with increased toxicity in the presence of muscle depletion	5-FU Fluoropyrimidine ± Oxaliplatin or Irinotecan (FOLFOX, FOLFIRI) Platinum (Cisplatin, Carboplatin) Epirubicin Taxane Sorafenib Sunitinib Safeni, vandetanib Pemetrexed Gemcitabine Vinorelbine Rituximab, cyclophosphamide, doxorubicin, vincristine, prednisolone (R-CHOP)
Outcomes for patients treated with immunotherapy in the presence of malnutrition/muscle depletion	Low muscle mass associated with poor outcomes, including lower response rate and shorter duration of response, for patients treated with immunotherapy for advanced cancers [48,58] Cachexia independently associated with worse overall survival in patients with NSCLC treated with PD-1 or PD-L1 inhibitors [59,60,61,62,63]

^a^ Median prevalence >20%. 5-FU, fluorouracil; FOLFOX, folinic acid, fluorouracil, oxaliplatin; R-CHOP, rituximab, cyclophosphamide, hydroxydaunorubicin hydrochloride, vincristine, prednisone.

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
