# Peer review of "Oncology-Led Early Identification of Nutritional Risk: A Pragmatic, Evidence-Based Protocol (PRONTO)"

_cancers, 2023, doi:10.3390/cancers15020380_

Round 1

Reviewer 1 Report

This manuscript by Muscaritoli  et al., presents an evidence based protocol for rapid and routine identification of patients with or at risk of malnutrition and/or muscle depletion. Overall, I found this article excellent, and I commend the authors on this work; it is an important addition to the literature. The authors provide an exceptionally concise review and balanced assessment of malnutrition in cancer from a world authority in the field.

I really like the conceptualization of a rapid tool for early identification of patients at risk of malnutrition about to start antineoplastic therapy. There is a clear need for a standardised approach to evaluate and monitor nutritional status, in order to maximize treatment efficacy and minimize the risk of toxicity.   

I have a few questions/comments for the authors:

·       Line 285-286: For the evaluation of weight loss, “Have you unintentionally lost weight (5% to 10% or more) in the last 3–6 months/since our last consultation?” Who is to calculate the 5-10%? Are the oncologists expected to do this calculation? Does the tool provide guidelines for oncologists on calculating percentage weight loss?

·       The difference between PRONTO and other validated malnutrition screening tools would be a valuable addition the paper. The supplementary material is very important as it needs to be clear to the reader why this new tool is needed and how it differs to the established tools available- why do we need another malnutrition screening tool?  

·       Dietetic workforce constraints are acknowledged in the paper. However, action points for the PRONTO tool direct the oncologist to refer to a nutrition expert for screening/assessment. In the absence of a nutrition expert, if may be helpful if there were links provided in the tool/decision making algorithm to evidence based dietary advice, that the oncologist can signpost the patient to – e.g. a weblink to PRONTO website that collates evidence-based nutrition resources for both health care professionals and patients. 

Reviewer 2 Report

The PRONTO protocol is an important step in optimizing nutritional care for patients with cancer. However, my major concern is that while the PRONTO initiative was taken in the context of the Italian setting, the context-specificity is not clear enough throughout the manuscript. The protocol including its recommendations cannot be generalized to other country settings, while the authors state that the PRONTO protocol "is suitable for implementation in all oncology settings". 
In other countries, like in the Netherlands, nutritional care for patients with cancer is organized differently, in which dietitians are strongly involved (nutritional assessment and dietary counselling including advising (or even deciding) on the choice of nutritional intervention/strategy) already. Also in other countries, oncology nurses may have a more prominent role in the multidisciplinary team already, in which nurses already screen patients for risk of malnutrition, after which a patient with positive screening result is being referred to a dietitian. In the Netherlands, patient groups known to have very high risk of malnutrition are even being referred to the dietitian routinely at start of their radiatin or chemoradiation treatment, and are being monitored by the dietitian and nurse throughout their treatment and even after treatment. So while the current protocol is likely to improve nutritional care for patients with cancer in the Italian setting, it does not necessarily improve nutritional care in other settings. In fact, implementing this protocol in countries like the Netherlands would mean a step back in nutritional care for patients with cancer.
Please explicitly describe the context-specificity of the PRONTO protocol in the manuscript, so that it becomes clear that the protocol was developed in the context of the healthcare situation in Italy, and that the protocol canNOT necessarily be generalized to other countries.

Below please find my suggestions to further strengthen the manuscript:
- Abstract: According to EWGSOP2, low muscle strength is the primary criterion for sarcopenia. Therefore, I would suggest to replace “loss of muscle function” by “low muscle strength”, and subsequently replace “loss of muscle mass”by “low muscle mass”.

- Abstract: “Poor nutritional status may lead to cancer cachexia” should be the other way around: “cancer cachexia may lead to poor nutritional status”

- The role of the oncology nurse in identifying patients requiring nutritional support can be strengthened in the protocol and manuscript. In addition, while it is mentioned that ideally the patient with risk of malnutrition is referred to a dietitian, it could be more explicitly mentioned that more dietitians are needed in Italy. Throughout the manuscript, it is often mentioned that dietitians are not always available; it seems like a missed opportunity to not more powerfully mention that more dietitians are needed to optimize nutritional care for oncology patients.

 - In the Glossary of terms, a reference for the definition of sarcopenia should be given

- Lines 86-88: would recommend to include quality of life as outcome influenced by malnutrition as well, as quality of life is one of the most important outcome measures as experienced by patients themselves. Lines 103-104 seem to suggest that quality of life is more important to older patients than those at younger age, while malnutrition negatively impacts quality of life at all ages.

- Lines 113-114: it is suggested that GLIM is a tool in itself to evaluate nutritional status; this is not fully correct: GLIM aims to diagnose malnutrition, parallel to thoroughly evaluating nutritional status by existing tools/methods. Only when a tool is not being used already, the GLIM could serve as a tool in itself.

- Line 130-131: it is unclear why simple measures like anthropometry to evaluate muscle mass have not been considered or even incorporated in the PRONTO protocol, especially since the GLIM Muscle mass Working Group (Compher, JPEN 2022) recommends anthropometry as one of the methods to determine muscle mass within the GLIM approach.

- Line 151 and throughout the manuscript: please replace "risk for malnutrition" by "risk of malnutrition".

- Page 6: the resolution of the illustration should be improved

- Line 227: head and neck cancer is also a tumor type associated with low muscle mass, and in these patients low muscle mass is also associated with early termination of chemotherapy, for example demonstrated by Sealy et al. In Clin Nutr 2019. Would reommend to add head and neck cancer here.

- Line 235: the statement “However, all patients with cancer are at increased risk for malnutrition and low muscle mass regardless of age and tumour type.” seems overestimated. For example, patients with a skin tumor on the back is not necessarily at risk of malnutrition. Nuancing of this statement is needed.

- I would recommend to replace “dietary advice” by “dietary counselling” throughout the manuscript, to address the patient-centered approach.
